

# MDDeep-Ace: species-specific acetylation site prediction based on multi-domain adaptation

Yu Liu, Chaofan Ye, Can Lin, Kangkang Mao and Ming Zhu

School of Integrated Circuits, Anhui University, Hefei City, Anhui, China

## ABSTRACT

**Background**. Lysine post-translational modification (PTM) is pivotal in regulating diverse cellular processes, profoundly impacting protein structure and function. Over recent decades, numerous experimental techniques have advanced PTM site identification, significantly contributing to research progress. However, these methods are time-intensive and labor-intensive. Deep learning technologies have shown promise in predicting PTM sites, yet current approaches struggle with species-specific PTM site prediction.

**Methods**. We introduce MDDeep-Ace, a novel deep learning method based on multi-domain adaptation for predicting lysine acetylation sites. By integrating data from multiple species, MDDeep-Ace enhances the generalization of species-specific prediction models, improving predictive performance.

**Results**. Experimental findings illustrate that our proposed multi-domain adaptation approach significantly enhances prediction accuracy across multiple species, surpassing existing lysine acetylation site prediction tools.

## INTRODUCTION

Protein acetylation is a critical post-translational modification (PTM) in which lysine acetyltransferase transfers an acetyl group from a donor to the $\varepsilon$-amino group side chain of lysine, typically at the protein's N-terminus. Initially identified in histone lysine residues of eukaryotic organisms, this reversible and tightly regulated process is vital for maintaining protein structure and function (*Kim et al., 2006*; *Narita, Weinert & Choudhary, 2018*; *Graf et al., 2021*). For example, research has shown that histone acetylation modification inhibits the promoter activity, mRNA, and protein expression of ADRB2, offering novel insights into asthma prevention and treatment (*Sheikhpour et al., 2021*). Additionally, acetylation modification contributes to the biological mechanisms of plant photomorphogenesis, such as seed germination, chlorophyll synthesis, and hypocotyl elongation (*Le Roux et al., 2015*; *Ma et al., 2020*). Thus, a comprehensive understanding of acetylation mechanisms is essential for elucidating cellular activity patterns and guiding disease management strategies.

Over the past few decades, various experimental techniques have advanced PTM site identification, driving progress in PTM research (*Aponte et al., 2009*; *Bockus & Scofield,*

Corresponding author
Ming Zhu, zhuming@ahu.edu.cn

*2009*). However, these methods are labor-intensive and time-consuming. With the exponential growth of protein data, relying solely on experimental approaches to identify PTM sites has become increasingly impractical. Consequently, there is an urgent need for computational methods to efficiently predict PTM sites on a large scale. For instance, *Deng et al. (2016)* developed a Bayes discrimination-based method called PAIL for acetylation site prediction, achieving over 85% accuracy across various thresholds. Similarly, *Shao et al. (2012)* proposed a Bayesian feature extraction approach in conjunction with support vector mechanisms for constructing an acetylation site prediction model, achieving high accuracy in predicting human acetylation sites. Furthermore, ProAcePred, proposed by *Chen et al., (2018)* employs elastic net optimization to enhance feature extraction, significantly improving species-specific lysine acetylation site prediction.

Although traditional machine learning-based PTM prediction methods have made notable progress, their dependence on manually extracted sequence features limits their ability to uncover complex patterns in large datasets. Recent advancements in deep learning methods have proven to be highly beneficial in the realms of artificial intelligence and biomedicine (*Wu et al., 2020*; *Lin et al., 2022*; *Zhu et al., 2023*; *Zou et al., 2023*; *Zou et al., 2024*). Their strength lies in their ability to automatically identify intricate patterns within training samples, leading to the abstract characterization of samples and enhanced prediction capabilities. For instance, *Chen et al. (2019)* developed MUscADEL, a bidirectional long short-term memory (LSTM) recurrent network-based tool for predicting PTM sites, including glycosylation, methylation, and ubiquitination, with superior performance. Additionally, CapsNet (*Wang, Liang & Xu, 2019*), replaces the convolutional layer with a capsule layer in the MusiteDeep framework (*Wang et al., 2017*), achieving outstanding performance across various PTM sites. *Lai & Gao (2023)* developed a web server named Auto-Kla, a transformer-based model. This model matches or exceeds the performance of existing acetylation site prediction models.

Despite the success of deep learning, developing species-specific PTM site prediction models requires substantial labeled data, which is often limited for many species. To address the issue of limited data, transfer learning has emerged as a solution to this challenge. For example, *Li et al. (2020)* used transfer learning to predict protease-specific cleavage sites by pre-training on extensive protease family data and fine-tuning on smaller, specific datasets. Besides fine-tuning, domain adaptation has been successfully employed in species-specific PTM site prediction. For instance, *Liu et al. (2021)* introduced DeepTL-Ubi, a domain adaptation-based method for predicting species-specific ubiquitination sites, boosting performance by incorporating a species transfer loss function to align semantics across different species. Subsequently, *Liu, Wang & Xi (2022)* developed DeepDA-Ace, a novel approach, leveraging semantic adversarial learning to minimize domain discrepancies among species, improving performance across multiple species. More recently, a semi-supervised method has been proposed to enhance species-specific PTM site prediction, which utilizes unlabeled data to augment species-specific data. Despite the effectiveness of these strategies in enhancing species-specific PTM site prediction, these methods primarily rely on human data as the source domain, neglecting valuable data from other species.

To address this gap, we design a multi-domain adaptation approach, MDDeep-Ace, utilizing PTM sites of multiple species to learn more discriminative patterns, enhancing the performance of the models. It employs a convolutional neural network-long short-term network (CNN-LSTM) hybrid network to extract sequence features and introduce a dynamic domain difference adjustment loss to modulate the influence of source domains. Extensive experiments demonstrate that MDDeep-Ace outperforms species-specific models trained on single-source domain, highlighting the benefits of multi-domain adaptation. Meanwhile, compared with the state-of-the-art lysine acetylation site prediction tools, MDDeep-Ace also achieves a remarkable performance improvement, establishing it as a robust tool for species-specific PTM site prediction.

This study conducted extensive ablation and comparison experiments to validate three hypotheses: (1) Domain adaptation enhances prediction accuracy for species with limited data, with MDDeep-Ace achieving a 7% higher average area under the curve (AUC) across nine species compared to non-domain adaptation methods. (2) Multi-species source domain models outperform single-source (human-only) models, with MDDeep-Ace showing AUC improvements of 4.2% and 2.0% for *B. velezensis* and *A. thaliana*, respectively. (3) In comparison with existing lysine PTM prediction tools, we finds that the deep learning-based methods, particularly species-specific models like MDDeep-Ace, surpass traditional machine learning approaches and general models.

The key contributions of this work include: (1) Proposing a multi-domain adaptation method that leverages the PTM sites of multiple species to improve the model's ability to generalize across species. (2) Designing a hybrid neural network to extract critical features for efficient bioinformatics prediction of PTM sites. (3) Demonstrating through experiments that MDDeep-Ace outperforms existing lysine PTM prediction tools. The code and data for MDDeep-Ace are available on GitHub: https://github.com/Lxiaoyuleyuan/MDDeep-Ace.

## MATERIALS AND METHODS

### Dataset

We collected ten species of lysine acetylation sites from PLMD dataset (*Xu et al., 2017*), including 6,078 *H. sapiens* proteins, 3,645 *M. musculus* proteins, 2,960 *S. cerevisiae* proteins, 4,359 *R. norvegicus* proteins, 1,251 *S. japonicum* proteins, 231 *A. thaliana* proteins, 1,860 *E. coli* proteins, 1,146 *B. velezensis* proteins, 1,214 *P. falciparum* proteins, 336 *O. sativa* proteins. We utilized the CD-HIT tool (*Huang et al., 2010*) to cluster protein sequences for each species, filtering out homologous proteins exceeding a 40% similarity threshold. Negative samples were defined as lysine residues that were not experimentally verified as acetylation sites from the non-homologous proteins. Subsequently, 10% of both positive and negative samples per species were randomly allocated to an independent test dataset, with the remaining data used for training and validation. To address sample imbalance and prevent over-optimization, we balanced the dataset by equalizing the number of positive and negative samples (Table 1). Finally, protein sequences were encoded into numerical vectors using one-hot encoding, where each amino acid is represented by a vector with a

**Table 1   The AUC value of MDDeep-Ace and baseline methods.**

| Species | Species-Specific | | | Genaral |
| --- | --- | --- | --- | --- |
| | MDDeep-Ace | SSDA | SSDT | MDDeep-Ace (Uniform weight) |
| R. norvegicus | 0.764 | 0.749 | 0.729 | 0.756 |
| S. japonicum | 0.818 | 0.802 | 0.727 | 0.813 |
| S. cerevisiae | 0.808 | 0.783 | 0.738 | 0.803 |
| M. musculus | 0.772 | 0.761 | 0.731 | 0.766 |
| E. coli | 0.763 | 0.748 | 0.742 | 0.755 |
| B. velezensis | 0.847 | 0.805 | 0.741 | 0.825 |
| P. falciparum | 0.700 | 0.682 | 0.659 | 0.692 |
| O. sativa | 0.823 | 0.804 | 0.689 | 0.803 |
| A. thaliana | 0.813 | 0.793 | 0.713 | 0.804 |
| Average | 0.790 | 0.770 | 0.719 | 0.779 |

single '1' and all other elements as '0'. For example, for lysine (K), the one-hot encoding is "00000010000000000000", while for serine (S), it is "00000000000000100000". By following previous studies (*Jia et al., 2016*; *Liu et al., 2021*), symmetrical 31-residue windows, spanning from −15 to +15 with lysine at the center, were extracted from protein fragments to serve as training and testing samples. Meanwhile, we also conducted comparative experiments on different window sizes, demonstrating the rationality of choosing a length of 31. The relevant settings and results of the experiments are listed in the Table S1. To standardize fragment lengths, we used the placeholder amino acid 'X' for padding missing residues or replacing non-standard amino acids. The fragments were transformed into numerical vectors using one-hot encoding. Each fragment was encoded as a 31 × 21 feature, where 31 represents the sequence length and 21 the amino acid categories.

## MDDeep-Ace architecture

MDDeep-Ace is a novel multi-domain adaptation learning based species-specific lysine acetylation site prediction method, with its workflow illustrated in Fig. 1. We curated data from ten species, using nine as source domains when training a predictive model for a specific target species.

The training dataset consists of two components: N labeled source dataset $D^S = \{D^{S_1}, D^{S_2}, \ldots, D^{S_N}\}$. The categories across all source domains are identical. $D^{S_j} = \left\{x_i^{S_j}, y_i^{S_j}\right\}_{i=1}^{N_{S_j}}$, where $x$ denotes the one-hot encoding sequence feature of sample and $y$ denotes the label of sample, $x_i^{S_j}$ and $y_i^{S_j}$ denote sequence feature of $j$th source domain and corresponding label, $N_{S_j}$ denotes the number of sample of $j$th source domain; a labeled target dataset $D^T = \left\{x_i^T, y_i^T\right\}_{i=1}^{N_T}$, where $x_i^T$ and $y_i^T$ denote sequence feature and label of target domain, $N_T$ denotes the number of labeled target domain.

MDDeep-Ace employs a CNN-LSTM hybrid network $g$ to extract sequence features and map protein sequences into an embedding space, followed by a classifier $h$ for feature classification. The CNN layer, with 21 input channels and 128 output channels, uses a kernel
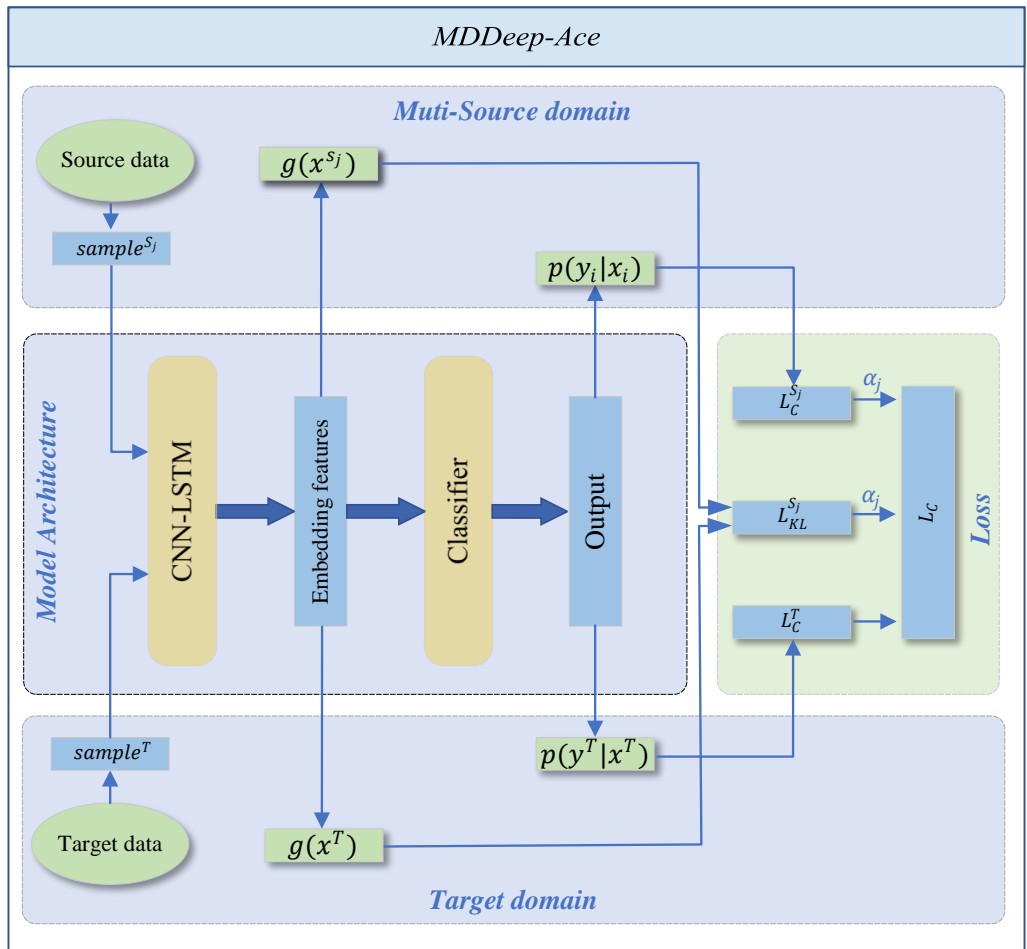

**Figure 1** **The overall framework of MDDeep-Ace.** The sample $s_j$ denotes the $j$-th data among multiple source domains, while the $sample^T$ represents the data in target domain.

size of 3 with a padding of 1 to capture local sequence features. The subsequent LSTM layer, with a hidden size of 128, processes the 128-channel CNN output to capture long-range dependencies. An average pooling layer (kernel size 2, stride 2) reduces dimensionality, enabling efficient extraction of both local and long-range features. The classifier applies a softmax operation to generate predictions.

$$p(y|x) = h(g(x:\theta); W). \tag{1}$$

The entire framework employs CNN-LSTM hybrid network as shared feature extractors across multiple domains. This shared structure allows data from different domains to be mapped to the same feature space, ensuring the extraction of domain-invariant features common to all fields. The domain alignment component minimizes feature distribution discrepancies between source and target domains through a domain discrepancy loss.

We employ Kullback–Leibler (KL) divergence to measure feature distribution distances between domains, calculating a dynamic adjustment factor to prioritize source domain

samples closer to the target domain while suppressing less similar ones. By differentially reducing inter-domain discrepancies, the framework better learns domain-invariant feature representations, thereby achieving more accurate recognition of target domain samples. To support the domain adaptation, the total loss comprises two parts: the classification loss of all source domains and the inter-domain discrepancy loss. By adjusting the weight proportions of each source domain within the total loss function, the influence of each source domain on the target domain is determined by the specific weight allocation. The total loss function is shown below:

$$L_{total} = L_C^T(g,h) + \alpha_j \left( L_C^{S_j}(g,h) + L_{KL}^{S_j}(g) \right) \tag{2}$$

where $L_C^T(\cdot)$ and $L_C^{S_j}(\cdot)$ represent the classification loss function applied to the target domain and the $j$th source domain, respectively, $L_{KL}^{S_j}(\cdot)$ denotes domain difference loss function for the $j$th source domain while $\alpha_j$ is the weight coefficient of the $j$th source domain in the loss function. The classification loss function is defined as follows:

$$L_c(g,h) = -\frac{1}{N} \sum_{i=1}^{Q} y_i \ln p(y_i|x_i) + (1-y_i) \ln (1 - p(y_i|x_i)), \tag{3}$$

where $Q$ is the number of samples. The domain difference loss function is defined as follows:

$$L_{KL}^{S_j} = \mathrm{E} \left( \sum_{k=1}^{M} g(x^T)_k \cdot \log \frac{g(x^T)_k}{g(x^{S_j})_k} \right) \tag{4}$$

where $\mathrm{E}(\cdot)$ denotes the mathematical expectation, $g(x)_k$ denotes the $k$th dimension feature of $g(x)$. The weight coefficient is computed by:

$$\alpha_j = \frac{\exp(D_j)}{\sum_{c=1}^{N} \exp(D_c)} \tag{5}$$

where $D_j$ denotes the KL divergence between the features of $j$th source domain and the target domain. Supplementary File S1 details the corresponding code for loss function calculations.

## Performance evaluation indicators

To evaluate MDDeep-Ace, we used five performance metrics: area under the receiver operating characteristic (ROC) curves (AUC), sensitivity (Sn), accuracy (Acc), precision (Pre) and F1 score. The formula of these metrics are as follows.

$$Acc = \frac{TP + TN}{TP + TN + FP + FN} \tag{6}$$

$$Sn = \frac{TP}{TP + FN} \tag{7}$$

$$Sp = \frac{TN}{TN + FP} \tag{8}$$

$$Pre = \frac{TP}{TP + FP} \tag{9}$$

$$F1 = \frac{2 \times Pre \times Sn}{Pre + Sn} \tag{10}$$

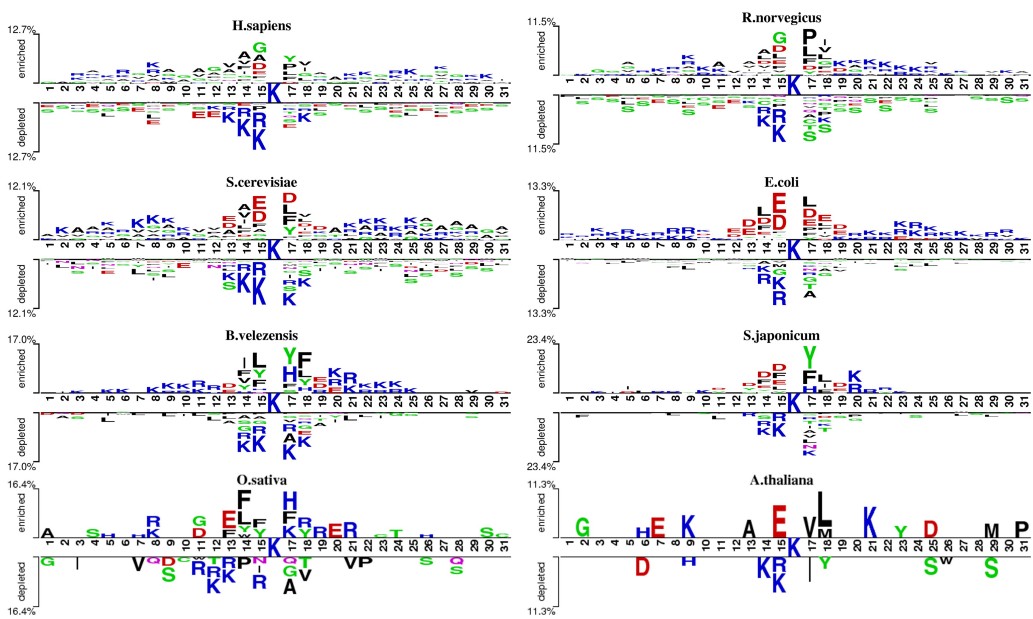

**Figure 2** **The two-sample-logo representation of position-specific residue composition surrounding the acetylation sites and non-acetylation sites.** The logo graphs are generated by the web server http://www.twosamplelogo.org/. Only residues significantly enriched or depleted ($t$-test, $P < 0.050$) flanking the centered acetylation sites (upstream 15 residues and downstream 15 residues) are shown.

where TP, TN, FP, FN represent true positives, true negatives, false positives and false negatives, respectively.

## RESULTS

### Sequence analysis

We analyzed the patterns of lysine acetylation using two-sample logo tool (Fig. 2). The analysis revealed distinct sequence differences across species. For example, Glycine/G residue is found enriched in position −1, −4 and +2 for *H. sapiens*, but not in *O. sativa* and *A. thaliana*. However, compared with *R. norvegicus* and *H. sapiens*, phenylalanine/F and tyrosine/Y residue prefer to enrich in position −1, −2 and +2 on *B. velezensis*. These findings underscore the need for species-specific prediction models. However, there still exist similarities between different species. For example, in most species, lysine (K) shows a significant depletion at diverse positions surrounding the acetylation site, both upstream and downstream. Meanwhile, the asparagine/D is enriched at −1 and +1 position on several species. Additionally, we also found that the degree of similarity varies among different species. For example, the types of amino acids enriched and depleted near acetylation sites in *S. cerevisiae* and *E. coli* are almost identical, whereas there are significant differences in the distribution of amino acids near acetylation sites between *S. cerevisiae* and *B. velezensis*. These observations highlight the potential of domain adaptation methods to leverage inter-species similarities, enhancing species-specific acetylation site prediction.

## Comparison with existing methods

We compare MDDeep-Ace with several existing acetylation site prediction methods including DeepDA-Ace (*Liu, Wang & Xi, 2022*), CapsNet (*Wang, Liang & Xu, 2019*), and PAIL (*Deng et al., 2016*) based on test dataset, and the construction approach of the test data set is shown as 2.1. In order to visualize the predictive performance, we show the ROC curves of different methods on multiple species in Fig. 3 and Fig. S1. From the Figure, we find that MDDeep-Ace, DeepDA-Ace and CapsNet achieved better performance than PAIL on all species. Take *S. cerevisiae* as an example, the AUC value of MDDeep-Ace, DeepDA-Ace and CapsNet are 0.808, 0.780 and 0.736 respectively, while PAIL only obtains 0.541 AUC value, demonstrating the superiority of deep learning-based methods. Besides, MDDeep-Ace and DeepDA-Ace outperform the other methods, suggesting that species-specific models have more advantages than general models. Meanwhile, it indicates that the transfer learning technique has an advantage over other deep learning methods in species-specific acetylation site prediction. For instance, DeepDA-Ace achieves 3.1%, 4.9%, 4.4% and 4.9% improvement compared to CapsNet on *S. japonicum*, *R. norvegicus*, *S. cerevisiae* and *M. musculus*. Additionally, our proposed MDDeep-Ace combines the benefits of transfer learning and multi-domain adaptation technique, and obtains the highest AUC value among existing methods on almost all species in test dataset. Taking *B. velezensis* as an example, compared to DeepDA-Ace, CapsNet and PAIL, the AUC of MDDeep-Ace has improved by 5.3%, 7.7% and 29.4%, respectively. As for *R. norvegicus*, MDDeep-Ace obtains 0.764 AUC value, while the corresponding AUC value of DeepDA-Ace, CapsNet and PAIL are 0.732, 0.683 and 0.538, respectively.

Additionally, to evaluate the performance of MDDeep-Ace more comprehensively, these four metrics are also considered by us: sensitivity (Sn), accuracy (Acc), precision (Pre) and F1 score at high specificity (Sp). We compared these metrics of the four methods at a specificity of 0.900, and the results are illustrated in Fig. 4 and Fig. S2. Clearly, MDDeep-Ace demonstrates superior performance across most metrics for all species. For example, on *S. cerevisiae*, MDDeep-Ace shows notable improvements: accuracy (Acc) increases by 5%, sensitivity (Sn) by 9.8%, precision (Pre) by 3.6%, and F1 score by 9.3% compared to PAIL. When compared to DeepDA-Ace, the improvements are also significant, with Acc, Sn, Pre, and F1 score increasing by 10.7%, 21%, 10.3%, and 21.8%, respectively. Moreover, compared with CapsNet, the gains are even more pronounced, with Sn, Acc, Pre, and F1 score increasing by 34.9%, 17.8%, 28.5%, and 40.4%, respectively. MDDeep-Ace not only outperforms these models in S.cerevisiae but also achieves competitive results across other species.

The superior performance of MDDeep-Ace can be attributed to two main factors. Firstly, MDDeep-Ace leverages multi-domain adaptation, effectively transferring acetylation knowledge from multiple species to enhance the model's predictive power. Secondly, MDDeep-Ace employs a CNN-LSTM hybrid network, which captures both local features and long-range dependencies, further boosting the model's predictive performance. Overall, MDDeep-Ace not only surpasses existing methods with respect to accuracy, sensitivity, precision, and F1 score, but also represents a significant advancement in

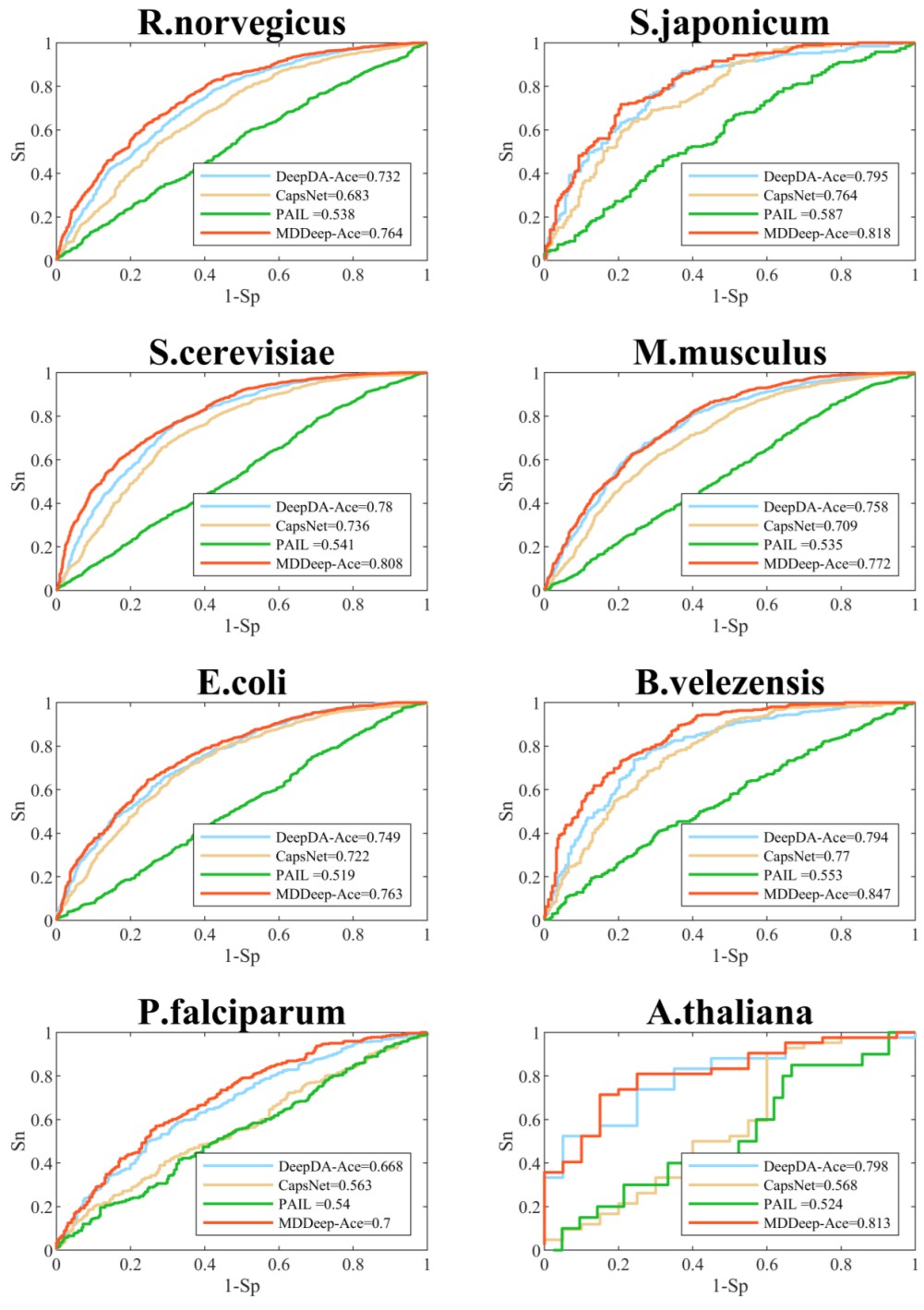

**Figure 3** Performance of ROC curves on *R. norvegicus, S. japonicum, S. cerevisiae, M. musculus, E. coli, B. velezensis, P. falciparum, A. thaliana.* The red lines represent the performance of MDDeep-Ace, the blue, orange and green lines represent the DeepDA-Ace, CapsNet and PAIL, respectively.

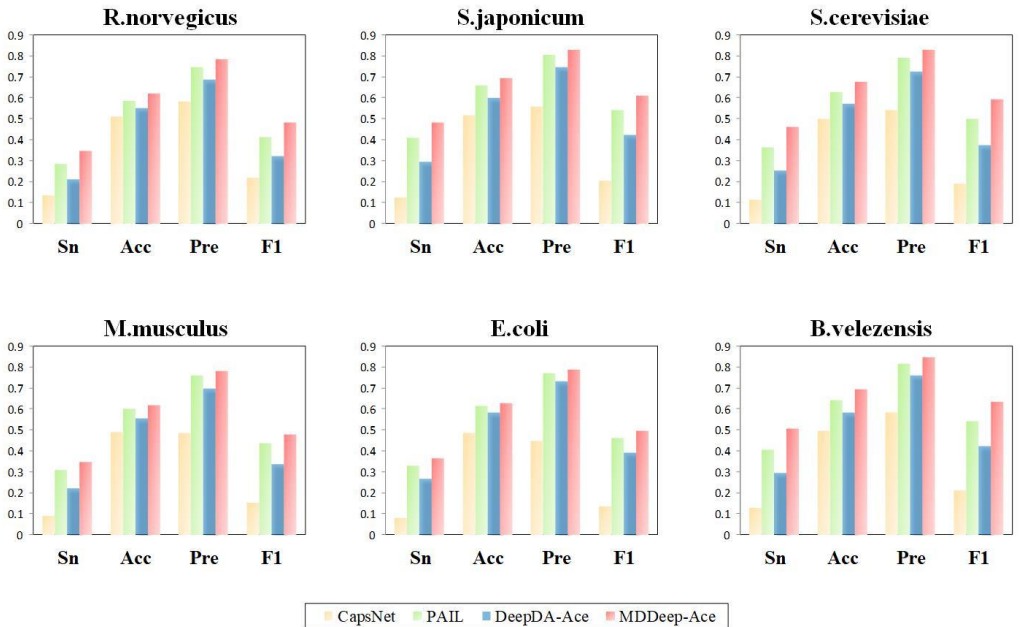

**Figure 4** The Sn, Acc, Pre, F1-score value comparison with different methods for *R. norvegicus, S. japonicum, S. cerevisiae, M. musculus, E. coli,* and *B. velezensis* at specificity of 0.900. The horizontal axis represents sensitivity, accuracy, precision and F1, respectively.

species-specific lysine acetylation site prediction, offering a robust and versatile tool for bioinformatics research.

## Ablation study

To validate the efficacy of the introduced multi-domain adaptation method, we compared MDDeep-Ace with three baseline models. (1) Species-specific direct training (SSDT) model: for each species, a supervised model is trained using only the labeled data specific to that species. (2) Species-specific domain adaptation (SSDA) model: this model leverages domain adaptation techniques, but uses only human data as source domain. It trains species-specific prediction models based on labeled data from both human and the target species. (3) Uniform-weight MDDeep-Ace model: to assess the impact of the domain weighting strategy, this model uses the same architecture as MDDeep-Ace but applies equal weighting to all source domains. The above three methods are all based on the CNN-LSTM hybrid network. The differences among them lie in either the distinct training data they utilize or the different training strategies they adopt. The results clearly demonstrate that the adaptive weighting mechanism in MDDeep-Ace plays a critical role in enhancing prediction accuracy.

Table 1 illustrates the performance of MDDeep-Ace compared to baseline methods, demonstrating that MDDeep-Ace generally outperforms the uniform-weight MDDeep-Ace in predicting PTM sites across various species. For instance, MDDeep-Ace achieves the highest AUC values across all species, reflecting its superior capability. However, it is also notable that the uniform-weight MDDeep-Ace performs comparably to the SSDA method in some species. For example, the AUC values of the uniform-weight MDDeep-Ace are only 0.5% and 0.7% higher than those of the SSDA method in *M. musculus* and

*R. norvegicus*, respectively. This can be attributed to the fact that human acetylation sites constitute a significant portion of the training data for the uniform-weight MDDeep-Ace, and the acetylation patterns of *M. musculus* and *R. norvegicus* are quite similar to those of humans. Conversely, the uniform-weight MDDeep-Ace performs poorly on *O. sativa* and *A. thaliana*, likely due to the considerable differences in acetylation patterns between these species and humans, making it challenging for the uniform-weight MDDeep-Ace to effectively capture species-specific patterns. These findings highlight the limitations of the uniform-weight MDDeep-Ace in accurately predicting PTM sites across all species, underscoring the necessity of establishing adaptive weight species-specific prediction models for enhanced performance.

Although SSDT is a species-specific method, it exhibits poor performance in some species such as *A. thaliana*, *O. sativa*, and *B. velezensis*, indicating the difficulty of building effective species-specific prediction models through direct training when training data is insufficient. To address this issue, domain adaptation technology, as employed in the SSDA method, transfers knowledge from extensive human PTM data to improve the prediction accuracy for other species. For instance, the SSDA method achieves AUC values of 0.793, 0.804, and 0.805 in *A. thaliana*, *O. sativa*, and *B. velezensis*, representing improvements of 8%, 11.5%, and 6.4% over SSDT. Moreover, with the integration of multiple species data, MDDeep-Ace further enhances prediction accuracy, achieving AUC values of 0.813, 0.823, and 0.847 in these species, respectively. In addition to these species, MDDeep-Ace also achieves the optimal performance across all other species, confirming its effectiveness in species-specific acetylation site prediction. The multi-domain adaptation strategy, combined with the incorporation of diverse species data, allows MDDeep-Ace to outperform existing methods.

Besides, we also compared the F1 scores, accuracy (Acc), sensitivity (Sn), and precision (Pre) for these methods, and the results are listed in Table 2. From the table, it is evident that the methods utilizing multi-domain adaptation technology achieved higher performance across several indicators. For example, in *R. norvegicus*, the Pre values of MDDeep-Ace and SSDA methods are 0.783 and 0.752, respectively, with MDDeep-Ace showing a 3.1% improvement over SSDA. Additionally, the Acc values of MDDeep-Ace and SSDA methods are 0.620 and 0.592, representing a 2.8% increase for MDDeep-Ace over SSDA. Moreover, compared to the uniform-weight MDDeep-Ace model, MDDeep-Ace demonstrates notable improvements. For instance, in *S. cerevisiae*, the Sn, Acc, Pre, and F1 score of MDDeep-Ace are 0.463, 0.677, 0.827, and 0.593, respectively, while the corresponding values for the uniform-weight MDDeep-Ace model are 0.431, 0.661, 0.817, and 0.565. This reflects a 3.2% increase in Sn, a 1.6% increase in Acc, a 1% increase in Pre, and a 2.8% improvement in F1 score. Among the domain adaptation-based methods, our proposed MDDeep-Ace shows clear advantages across most species. For example, in *B. velezensis*, the Pre, Acc, F1, and Sn of MDDeep-Ace are 0.846, 0.694, 0.634, and 0.507, respectively, whereas the SSDA method only achieves 0.774, 0.594, 0.449, and 0.316. This represents improvements of 7.2%, 10%, 18.5%, and 19.1% in Pre, Acc, F1, and Sn, respectively, when using MDDeep-Ace. These results further support the conclusions drawn from the comparison using AUC values.

**Table 2  Performance comparison of MDDeep-Ace with baseline methods including direct training, fine tuning and combined method on nine different species.**

| Species | Method | Pre | Sn | Acc | Pre | F1-score |
|---|---|---|---|---|---|---|
| | | **Sp = 0.900** | | | | |
| R. norvegicus | general | MDDeep-Ace (Uniform weight) | 0.333 | 0.613 | 0.776 | 0.466 |
| | species-specific | MDDeep-Ace | 0.348 | 0.620 | 0.783 | 0.481 |
| | | SSDA | 0.292 | 0.592 | 0.752 | 0.421 |
| | | SSDT | 0.295 | 0.593 | 0.754 | 0.424 |
| S. japonicum | general | MDDeep-Ace (Uniform weight) | 0.419 | 0.662 | 0.808 | 0.551 |
| | species-specific | MDDeep-Ace | 0.482 | 0.694 | 0.829 | 0.609 |
| | | SSDA | 0.382 | 0.644 | 0.793 | 0.516 |
| | | SSDT | 0.272 | 0.590 | 0.732 | 0.397 |
| S. cerevisiae | general | MDDeep-Ace (Uniform weight) | 0.431 | 0.661 | 0.817 | 0.565 |
| | species-specific | MDDeep-Ace | 0.463 | 0.677 | 0.827 | 0.593 |
| | | SSDA | 0.325 | 0.607 | 0.771 | 0.457 |
| | | SSDT | 0.340 | 0.615 | 0.779 | 0.474 |
| M. musculus | general | MDDeep-Ace (Uniform weight) | 0.327 | 0.608 | 0.771 | 0.459 |
| | species-specific | MDDeep-Ace | 0.346 | 0.618 | 0.781 | 0.479 |
| | | SSDA | 0.321 | 0.605 | 0.768 | 0.452 |
| | | SSDT | 0.286 | 0.587 | 0.747 | 0.413 |
| E. coli | general | MDDeep-Ace (Uniform weight) | 0.322 | 0.609 | 0.766 | 0.453 |
| | species-specific | MDDeep-Ace | 0.363 | 0.629 | 0.787 | 0.497 |
| | | SSDA | 0.279 | 0.587 | 0.739 | 0.405 |
| | | SSDT | 0.339 | 0.617 | 0.775 | 0.472 |
| B. velezensis | general | MDDeep-Ace (Uniform weight) | 0.493 | 0.687 | 0.843 | 0.622 |
| | species-specific | MDDeep-Ace | 0.507 | 0.694 | 0.846 | 0.634 |
| | | SSDA | 0.316 | 0.594 | 0.774 | 0.449 |
| | | SSDT | 0.316 | 0.594 | 0.774 | 0.449 |
| P. falciparum | general | MDDeep-Ace (Uniform weight) | 0.283 | 0.577 | 0.758 | 0.413 |
| | species-specific | MDDeep-Ace | 0.259 | 0.564 | 0.741 | 0.383 |
| | | SSDA | 0.215 | 0.541 | 0.704 | 0.329 |
| | | SSDT | 0.199 | 0.533 | 0.688 | 0.309 |
| O. sativa | general | MDDeep-Ace (Uniform weight) | 0.500 | 0.685 | 0.857 | 0.632 |
| | species-specific | MDDeep-Ace | 0.458 | 0.663 | 0.846 | 0.595 |
| | | SSDA | 0.417 | 0.640 | 0.833 | 0.556 |
| | | SSDT | 0.208 | 0.528 | 0.714 | 0.323 |
| A. thaliana | general | MDDeep-Ace (Uniform weight) | 0.357 | 0.532 | 0.882 | 0.508 |
| | species-specific | MDDeep-Ace | 0.405 | 0.565 | 0.895 | 0.557 |
| | | SSDA | 0.381 | 0.548 | 0.889 | 0.533 |
| | | SSDT | 0.262 | 0.468 | 0.846 | 0.400 |

## DISCUSSION

Despite the strong results achieved by MDDeep-Ace in lysine PTM site prediction, there remains potential for further improvement through refinements in training methods and network architecture. For example, recent research by *Lai & Gao (2023)* demonstrated

that natural language processing models like transformers can be effectively applied to acetylation site prediction, offering high robustness and generalization ability. In future work, we aim to integrate transformer architecture into MDDeep-Ace to enhance feature extraction. Additionally, there are millions of lysine sites in protein sequences, some of which may not be confirmed as acetylated, and which can be used as unlabeled data to pre-train models. Therefore, we plan to explore leveraging this unlabeled data to transfer knowledge, thereby improving PTM site prediction accuracy for target species.

## CONCLUSIONS

Lysine PTMs play a critical role in controlling protein functions and numerous biological processes. Identifying lysine PTM sites is crucial for advancing our understanding of the molecular mechanisms underlying these modifications. However, there remains significant room for improvement in species-specific PTM site prediction using existing methods. In this study, we introduce MDDeep-Ace, a multi-domain adaptation method, designed to predict species-specific PTM sites with high accuracy. MDDeep-Ace consistently outperforms the state-of-the-art PTM site prediction tools across multiple species.

The exceptional performance of MDDeep-Ace can be attributed to several key factors: (1) Unlike existing PTM site prediction tools such as MUscADEL and CapsNet, MDDeep-Ace leverages domain adaptation techniques, effectively utilizing other species knowledge to enhance PTM prediction in small sample species. This approach addresses the limitations of models that rely solely on species-specific data. (2) Compared to transfer learning-based methods like DeepDA-Ace and DeepTL-Ubi, MDDeep-Ace integrates multiple species data, significantly boosting model generalization. (3) MDDeep-Ace employs a CNN-LSTM hybrid network, which excels at capturing both local features and long-range dependencies within protein sequences, further enhancing predictive accuracy. (4) A novel adaptive weighting mechanism dynamically prioritizes source domains based on their similarity to the target species, optimizing domain adaptation. These advancements position MDDeep-Ace as a powerful tool for species-specific lysine PTM prediction, offering valuable insights for developing novel computational approaches in bioinformatics.

### Funding

This research was funded by the National Natural Science Foundation of China, grant number 62202003. The funders had no role in study design, data collection and analysis, decision to publish, or preparation of the manuscript.

### Grant Disclosures

The following grant information was disclosed by the authors:
National Natural Science Foundation of China: 62202003.

### Competing Interests

The authors declare there are no competing interests.

## Author Contributions

- Yu Liu conceived and designed the experiments, authored or reviewed drafts of the article, and approved the final draft.
- Chaofan Ye analyzed the data, authored or reviewed drafts of the article, and approved the final draft.
- Can Lin performed the experiments, authored or reviewed drafts of the article, and approved the final draft.
- Kangkang Mao analyzed the data, prepared figures and/or tables, and approved the final draft.
- Ming Zhu analyzed the data, authored or reviewed drafts of the article, and approved the final draft.

## Data Availability

Code and raw data are available in the Supplemental Files.

## Supplemental Information

Supplemental information for this article can be found online at http://dx.doi.org/10.7717/peerj.19649#supplemental-information.

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
