# Peer review of "MDDeep-Ace: species-specific acetylation site prediction based on multi-domain adaptation"

_PeerJ, doi:10.7717/peerj.19649_

## Round 0.1 · original submission · Major Revisions

Please revise the manuscript by following the reviewers' comments.

·

Basic reporting

1. line 117: A reference for the PLMD dataset is needed.
2. line 200: It would be better to define these metrics in the Methods rather than the Results.
3. lines 233-243: the text in this paragraph is repetitive

Experimental design

1. A better description of the test dataset (line 201) is needed. Was this dataset independent of the training data used to build MDDeepAce? Was it independent of the training data used for the models MDDeepAce is being compared to?
2. How did the authors settle on using a 31 amino acid window around the acetylation site? Were different window sizes tested?
3. lines 249-254: A more extensive description of the three comparison models used in the ablation study would be helpful. (This could be added to the Methods.) For example, it isn't clear how model #2 is different from MDDeep-Ace.

Minor
4. line 145: define x and y

Validity of the findings

no comment

Additional comments

In this work, the authors address an important problem--prediction of lysine acetylation sites in multiple species, including species where there is limited experimental data on acetylation. For each target species, their model--MDDeepAce--uses data from multiple other species and captures common features of acetylation across species as well as species-specific features. The performance of MDDeepAce exceeds performance of other state-of-the-art models. The work is generally good, but the authors should address the above points before publication.

Reviewer 2 ·

Basic reporting

Review of MDDeep-Ace: Species-Specific Acetylation Site Prediction Based on Multi-Domain Adaptation

Overall Assessment:

The manuscript presents a method called MDDeep-Ace, a multi-domain adaptation technique for predicting species-specific lysine acetylation sites. The authors address the problem of limited data in predicting Post-Translational Modification (PTM) sites by leveraging data from multiple species. The method seems promising, showing improved accuracy compared to existing tools. However, there are areas where the manuscript and associated code could be strengthened.

Detailed Review:

1. Manuscript (peerj-reviewing-114276-v0.pdf):

BASIC REPORTING:

Clarity and English: The English is generally clear but needs improvement. There are instances of awkward phrasing and grammatical errors that affect readability. (e.g., "adaptative" instead of "adaptive," some unclear sentence structures). It would benefit from professional editing.  
Introduction and Background: The introduction provides sufficient background on protein acetylation and the challenges in predicting PTM sites. The literature review is relevant, covering traditional machine learning and recent deep learning approaches.  
Structure: The manuscript follows a standard structure (Abstract, Introduction, Materials and Methods, Results, Discussion, Conclusion). It conforms to PeerJ standards.  
Figures:
Figures are generally well-labeled, but their resolution could be improved for better clarity, especially in Figure 2.  
Figure 1 clearly illustrates the MDDeep-Ace framework.  
The ROC curves in Figure 3 and Supplemental Figure S1 are appropriate for showing the predictive performance.  
Figure 4 and Supplemental Figure S2 effectively compare the performance metrics.  
Tables: Tables 1 and 2 are clear and informative, presenting the quantitative results effectively.  
Raw Data: The authors have supplied raw data files, which is commendable.

Experimental design

EXPERIMENTAL DESIGN:

Research Question: The research question (improving species-specific PTM site prediction) is well-defined, relevant, and significant.  
Knowledge Gap: The authors clearly state how their research fills the knowledge gap: addressing the inadequacy of current deep learning approaches in predicting species-specific PTM sites and the limitations of existing transfer learning methods that don't utilize multi-species data.  
Methods: The methods are described in sufficient detail for replication. The use of a hybrid CNN-LSTM network and the dynamic domain difference adjustment loss is well explained.  
Datasets: The datasets used are appropriate and clearly described, including the data processing steps.

Validity of the findings

VALIDITY OF THE FINDINGS:

Data Robustness: The data appears to be robust. The balancing of positive and negative samples is a good practice.  
Statistical Soundness: The statistical methods used for evaluation (AUC, accuracy, sensitivity, precision, F1 score) are standard and appropriate.  
Controls: The ablation study provides good internal controls, demonstrating the importance of multi-domain adaptation and the adaptive weighting mechanism.  
Conclusions: The conclusions are generally well-supported by the results and are appropriately limited. The authors acknowledge the potential for further improvement, which is a positive sign.

Additional comments

Code (MDDeep-Ace folder):

Organization: The code is reasonably organized into different Python files (e.g., main_stage1.py, main_stage2.py, datasets.py, model/).
Clarity: The code could benefit from more comments to explain the purpose of different sections and variables. Variable names are generally adequate.
Functionality:
main_stage1.py: This script seems to handle the pre-training stage. It includes the model definition, training loop, and evaluation.
main_stage2.py: This script implements the multi-domain adaptation. It loads the pre-trained model and fine-tunes it. The KL divergence loss and dynamic weight adjustment are implemented here.
datasets.py: This file defines the data loading and augmentation procedures. The dataset_stage1 and dataset_stage2 classes are crucial for handling the different training strategies.
model/: This directory contains the model architectures (CNN-LSTM, Classifiers). The densenet.py file requires careful review to ensure the network structure aligns with the manuscript description.
data_process.py and data_general.py: These scripts handle data preprocessing, including one-hot encoding and data splitting.
Dependencies: The code uses common libraries like PyTorch, NumPy, and scikit-learn. The dependencies should be explicitly listed in a requirements.txt file.
Reproducibility: To enhance reproducibility, the authors should:
Provide clear instructions on how to run the code (e.g., command-line arguments, environment setup).
Include example data or specify how to obtain the data.
Consider using a version control system (like Git) to track changes and provide a stable release.
Suitability for PeerJ Publication:

The manuscript, with revisions, is potentially suitable for publication in PeerJ. The method addresses a relevant problem and shows promising results.
The code is a valuable contribution, but improvements in clarity, documentation, and reproducibility are necessary.
Recommendations for Revision:

Manuscript:

Language Editing: Thoroughly edit the English for clarity and grammatical correctness.
Figure Quality: Improve the resolution and clarity of all figures.
Introduction: Add a stronger concluding paragraph to the Introduction that clearly states the hypotheses being tested and the specific contributions of the work.
Methods Clarity:
Incorporate more explicit connections between the equations and the code implementation. For example, explicitly state which lines of code correspond to which parts of the loss function calculation.
Add more detail to the description of the CNN-LSTM architecture. Specify the number of layers, kernel sizes, filter sizes, etc., either in the main text or the supplementary material.
Discussion: Expand the discussion to address potential limitations of the study and future research directions in more detail.
References: Ensure all references are correctly formatted and complete.
Code:

Code Comments: Add comprehensive comments to the code, explaining the logic and purpose of different parts.
Documentation: Provide a README file with clear instructions on how to run the code, install dependencies, and prepare the data.
Dependencies File: Include a requirements.txt file to list all Python packages required to run the code.
Example Data: Include a small sample dataset for testing the code or provide a clear link and instructions on how to download the full dataset.
Version Control: Consider using Git and GitHub/GitLab to manage the code and provide versioning.
Parameter Explanation: Explain the meaning of all command-line arguments and key parameters used in the scripts.
Output Description: Clearly describe the format and content of the output files generated by the code.
Modularization: Further modularize the code into functions and classes to improve readability and maintainability.
By addressing these points, the authors can significantly improve their manuscript and code, making it a stronger and more impactful contribution to the field.

---

## Round 0.2 · Minor Revisions

Please revise the manuscript accordingly.

**Language Note:** The review process has identified that the English language must be improved. PeerJ can provide language editing services - please contact us at [email protected] for pricing (be sure to provide your manuscript number and title). Alternatively, you should make your own arrangements to improve the language quality and provide details in your response letter. – PeerJ Staff

·

Basic reporting

Check line 289: It looks like a sentence from the response document was inadvertently included in the manuscript. Also, the manuscript would benefit from smoothing of the grammar, particularly in the sections that were added in the revision.

Experimental design

No comment.

Validity of the findings

No comment.

Additional comments

No comment.

Reviewer 2 ·

Basic reporting

publish

Experimental design

publish

Validity of the findings

publish

---

## Round 0.3 · accepted · Accept

The manuscript can now be accepted for publication.

·

Basic reporting

No comment.

Experimental design

No comment.

Validity of the findings

No comment.

Additional comments

Thank you for addressing my comments. Everything looks good. I support publication.